# TEMPORAL AND OBJECT QUANTIFICATION NETS

## ABSTRACT

We aim to learn generalizable representations for complex activities by quantifying over both entities and time, as in "the kicker is behind all the other players," or "the player controls the ball until it moves toward the goal." Such a structural inductive bias of object relations, object quantification, and temporal orders will enable the learned representation to generalize to situations with varying numbers of agents, objects, and time courses. In this paper, we present *Temporal and Object Quantification Nets* (TOQ-Nets), which provide such structural inductive bias for learning composable action concepts from time sequences that describe the properties and relations of multiple entities. We evaluate TOQ-Nets on two benchmarks: trajectory-based soccer event detection, and 6D pose-based manipulation concept learning. We demonstrate that TOQ-Nets can generalize from small amounts of data to scenarios where there are more agents and objects than were present during training. The learned concepts are also robust with respect to temporally warped sequences, and easily transfer to other prediction tasks in a similar domain.

## 1 INTRODUCTION

When watching a soccer match (Fig. 1), we see more than just players and a ball moving around. Rather, we see events and actions in terms of high-level concepts, including relations between agents and objects. For example, people can easily recognize when one player has control of the ball, or when a player *passed* the ball to another player. This cognitive act is effortless, intuitive, and fast.

Machines can recognize actions, too, but generally based on limited windows of space and time, and with weak generalization. Consider the variety of complicated spatio-temporal trajectories that *passing* can refer to (Fig. 1), and how an intelligent agent could learn this. The act of passing does not seem to be about the pixel-level specifics of the spatio-temporal trajectory. Rather, a pass is a high-level action-concept that is composed of other concepts, such as *possession* (A pass begins with one player in possession of the ball and ends with another player in possession) or *kick*. The concept of passing can itself be re-used in a compositional way, for example to distinguish between a *short pass* and a *long pass*.

We propose Temporal and Object Quantification Nets (TOQ-Nets), structured neural networks that learn to describe complex activities by quantifying over both entities and time. TOQ-Nets are motivated by the way in which humans perceive actions in terms of the properties of and relations between agents and objects, and the sequential structure of events (Zacks et al., 2007; Stränger & Hommel, 1996).

A TOQ-Net is a multi-layer neural network whose inputs are the properties of agents and objects and their relationships in a scene, which may change over time. In a soccer game, these inputs might be the 3D position of each player and the ball. Each layer in the TOQ-Net performs either *object* or *temporal* quantification, which can emulate and realize disjunctive and conjunctive quantification over the properties and relationships between entities, as in "the kicker is behind *all* the other players", as well as quantification over time, as in "the player controls the ball *until* it moves fast towards the goal." The key idea of TOQ-Nets is to use tensors to represent the relational features between agents and objects (e.g., the player *controls* the ball, and the ball is *moving fast*), and to use tensor pooling operations over different dimensions to realize temporal and object quantifiers (*all* and *until*). Thus, by stacking these object and temporal quantification operations, TOQ-Nets can learn to construct higher-level concepts of actions based on the relations between entities over time, starting from low-level position and velocity input and supervised with only high-level class labels.

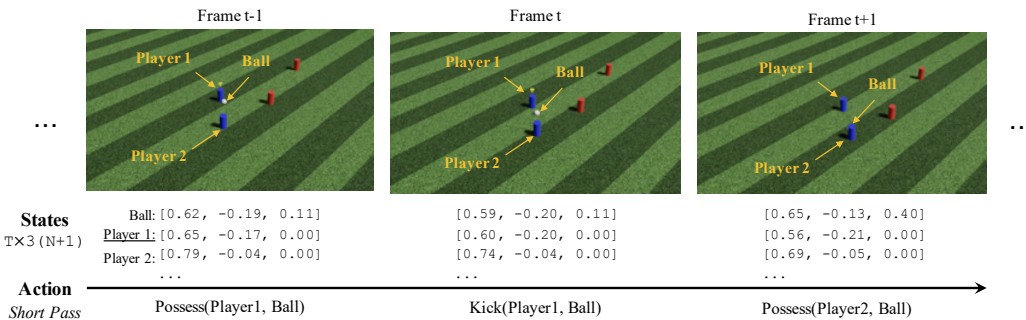

Figure 1: Illustrative example of action recognition in a soccer simulator. The action concept *short pass* is composed of other concepts, such as *possession* and *kick*.

We evaluate TOQ-Nets on two perceptually and conceptually different benchmarks for action recognition: trajectory-based soccer event detection and 6D pose-based manipulation concept learning. We show that the TOQ-Net makes several important contributions. First, TOQ-Nets outperform both convolutional and recurrent baselines for modeling relational-temporal concepts across both benchmarks. Second, by exploiting the temporal-relational features learned through supervised learning, TOQ-Nets achieve strong few-shot generalization to novel actions. Finally, TOQ-Nets exhibit strong generalization to scenarios where there are more agents and objects than were present during training. They are also robust w.r.t. time warped input trajectories. Meanwhile, the learned concepts can also be easily transferred to other prediction tasks in a similar domain.

## 2 RELATED WORK

**Action concept representations and learning.**   TOQ-Nets impose a structural inductive bias that describes actions with quantification over objects (*every* entity ..., there *exists* an entity ...)  and quantification over times (an event happens *until* some time and *then* some other event begins), which is motivated by first-order and linear temporal logics (Pnueli, 1977). Such representations have been studied for analyzing sporting events (Intille & Bobick, 1999; 2001) and daily actions (Tran & Davis, 2008; Brendel et al., 2011) using logic-based reasoning frameworks. However, these frameworks require extra knowledge to annotate relationships between low-level, primitive actions and complex ones. By contrast, TOQ-Nets enable end-to-end learning of complex action descriptions with only high-level action-class labels.

Cognitive science has long recognized the importance of structural representations of actions and events in human reasoning, including in language, memory, perception, and development (see e.g. Stränger & Hommel, 1996; Zacks et al., 2001; 2007; Pinker, 2007; Baldwin et al., 2001). In this paper, we focus on the role of object and temporal quantification in learning action representations.

**Temporal and relational reasoning.**   This paper is also related to work on using neural networks and other data-driven models for modeling temporal structure. Early work includes ADL description languages (Intille & Bobick, 1999; Zhuo et al., 2019), hidden Markov models (Tang et al., 2012), and and-or graphs (Gupta et al., 2009; Tang et al., 2013). These models need human-annotated action descriptions (e.g., *pick up* $x$ means a state transition from not holding $x$ to holding $x$) and special-purpose inference algorithms such as graph structure learning algorithms. In contrast, TOQ-Nets have an end-to-end design and can be integrated with arbitrary differentiable modules such as convolutional neural networks. People have also used structural representations to model object-centric temporal concepts with temporal graph convolution networks (Yan et al., 2018; Materzynska et al., 2020; Wang & Gupta, 2018; Ji et al., 2020). Meanwhile, Deng et al. (2016) proposed to integrate graph neural networks with RNNs by replacing the message aggregation step in GNN propagation with RNNs to capture temporal information. Bialkowski et al. (2014); Ibrahim et al. (2016)) have proposed to build a personal-level and a group-level feature extractor for group activity recognition. The high-level idea is to use RNNs to encode per-person features across the video and use another RNN to combine features of individual persons into a group feature for every frame. StagNet Qi et al. (2018) and Spatial-Temporal Interaction Networks (Materzynska et al., 2020) combines graph neural networks with RNNs and spatial-temporal attention models to model temporal information.  TOQ-Nets use a similar object-centric relational representation, but different models for temporal structures. There

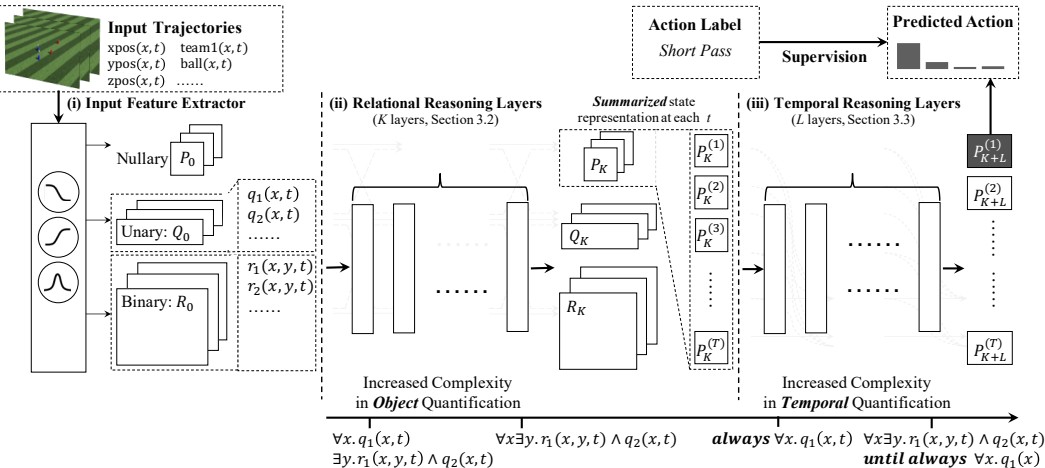

Figure 2: A TOQ-Net contains three modules: (i) an input feature extractor, (ii) relational reasoning layers, and (iii) temporal reasoning layers. To illustrate the model's representational power, we show logical forms of increasing complexity that can be realized by stacking multiple layers.

are also alternative models for temporal structures from pixels, such as convolutions (Carreira & Zisserman, 2017; Ji et al., 2020), recurrent networks (Dong et al., 2019), attention (Wang et al., 2018), or segmentation consensus and its variants (Wang et al., 2016; Zhou et al., 2018). In contrast to these methods, TOQ-Nets works on an object-centric temporal-relational feature space, and uses the proposed temporal quantification operations to model the sequential structures of events.

## 3 TEMPORAL AND OBJECT QUANTIFICATION NETS

TOQ-Nets, shown in Fig. 2, are multi-layer neural networks with a structural inductive bias for learning composable action concepts using object and temporal quantification. The input to a TOQ-Net is a tensor representation of the properties of all entities at each moment in time. For example, in a soccer game, the input encodes the position of each player and the ball, as well as the team membership of each player. The output is a label for the action performed by a target entity.

The first layer of a TOQ-Net (Fig. 2 (i)) extracts temporal features for each entity with an *input feature extractor* that focuses on entity features within a fixed and local time window. These features may be computed by a convolutional neural network, such as STGCN (Yan et al., 2018) or a bank of parametric feature templates, such as thresholding the velocity of an entity: whether the velocity of an entity $x$ is greater than a learnable parameter $\theta$ at time $t$. The output of this step is thus a collection of nullary, unary, and binary relational features over time for all entities. Throughout the paper we will assume that all output tensors of this layer are binary-valued, although the one can easily extend this feature extractor to any real-valued function. This input feature extractor is usually task- and dataset-specific and is not the main focus of this paper.

Second, the temporal-relational features go through several *relational reasoning* layers, which we present in detail in Section 3.2. In short, each layer at this step performs linear transformations, sigmoid activation, and *object quantification* operations. The linear and sigmoid functions allow the network to realize Boolean logical forms, and object quantification operators can realize quantifiers. With more relational reasoning layers, the network can realize logical forms with more quantifiers, as illustrated in Fig. 2. All operators in these layers are performed for all time steps in parallel.

Next, the relational reasoning layers summarize each event at a particular time step as a nullary feature that is passed to the to the *temporal reasoning layer*, as detailed in Section 3.3. Each layer at this step performs linear transformations, sigmoid activation, and *temporal quantification*. They allow the model to realize logical forms in a simple linear temporal logic (Pnueli, 1977), such as *always* $\exists x.\, q_1(x) \land q_2(x)$. This is semantically equivalent to the temporally quantified expression $\forall t.\, \exists x.\, q_1(x,t) \land q_2(x,t)$. Similar to the relational reasoning layers, adding more temporal reasoning layers enables the network to realize logical forms with more deeply nested temporal quantifiers.

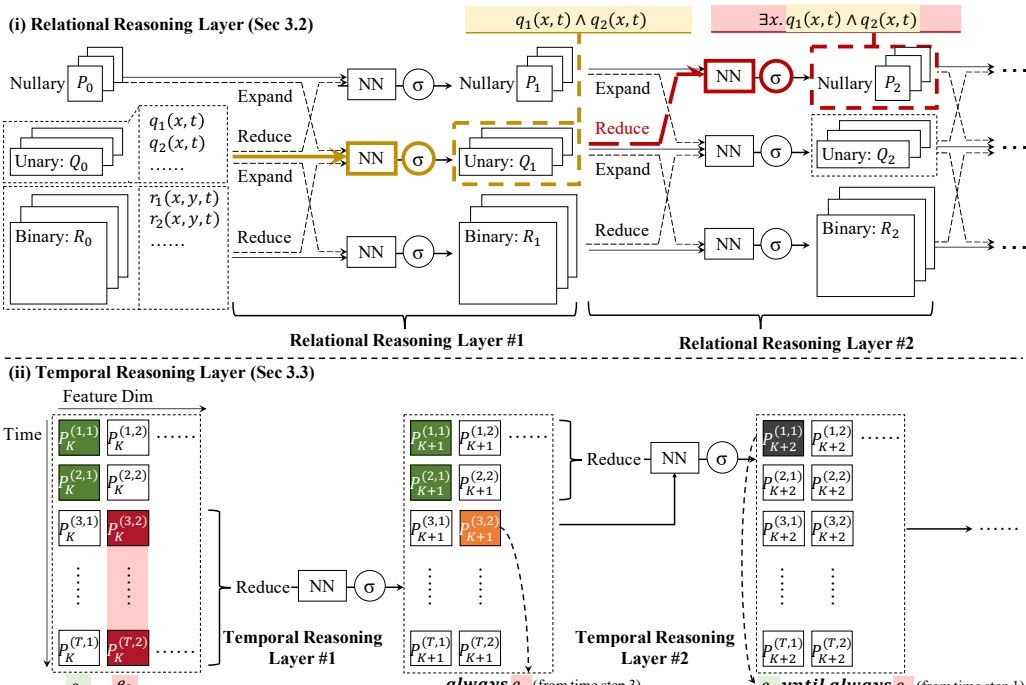

Figure 3: Illustration of (i) relational reasoning layers and (ii) temporal reasoning layers. We provide two illustrative running traces. (i) The first relational reasoning layer takes unary predicates $q_1$ and $q_2$ as input and its output $Q_1$ is able to represent $q_1 \wedge q_2$. The $\max(Q_1, dim = 0)$ in layer 2 can represent $\exists x.\ q_1(x, t) \wedge q_2(x, t)$. (ii) Assume $P_K$ encodes the occurance of events $e_1$ and $e_2$ at each time step. The first temporal reasoning layer can realize *always* $e_2$ with a temporal pooling from time step 3 to time step $T$. In the second temporal reasoning layer, the temporal pooling summarizes that $e_1$ holds true from time step 1 to 2. Thus, the NN should be able to realize $e_1$ *until* (*always* $e_2$).

In the last layer, all object and time information is projected into a representation in the initial time step, which summarizes what takes place in the entire trajectory (e.g., the event "the kicker eventually scores" may be recognized at the first time step). We feed this representation into a final softmax unit, and get classification probabilities for the event that the network is trained to detect.

**Remark.** It is important to understand the representational power of this model. The *input transformation layer* can be seen as defining basic predicates and relations that will be useful for defining more complex concepts, but no specific predicates or relations are built into it. The learned outputs might be properties that humans can interpret, such as "moving fast" or "far from." The relational reasoning layers can build quantified expressions over these basic properties and relations, and might construct expressions that could be interpretable as "the player is close to the ball." Finally, the temporal reasoning layer can represent temporal operations applied to these complex expressions, such as "the player is close to the ball until the ball moves with high speed." Critically, *none* of the symbolic properties or predicates are hand defined—they are all constructed by the initial layer in order to enable the network to express the concept it is being trained on.

Compared with the expressiveness of first-order linear temporal logic, TOQ-Nets only work in finite universes: all quantifiers only have finite interpretations. Meanwhile, the depth of the logical forms that it can learn is fixed. However, our goal is not to fully replicate temporal logic, but to bring ideas of object and temporal quantification into neural networks, and to use them as structural inductive biases to build models that generalize better from small amounts of data to situations with varying numbers of agents, objects, and time courses.

## 3.1 TEMPORAL-RELATIONAL FEATURE REPRESENTATION

TOQ-Nets use tensors to represent object-centric and relational features over time. The input to the network as well as the feature output of intermediate layers is represented as a tuple of three tensors.

Specifically, we use a vector of dimension $D_0$ to represent aspects of the state that apply to or depend on all entities at a specific time step $t$. We use a matrix of shape $N \times D_1$ to represent the state of each entity at time $t$, where $N$ is the number of entities and $D_1$ is the hidden dimension size. Similarly, we use a tensor of shape $N \times N \times D_2$ to represent a pairwise relation between each pair of entities at time step $t$. As a concrete example, illustrated in Fig. 2, the number of entities $N$ is the total number of players plus one (the ball). For each entity $x$ and each time step, the inputs are their 3D position, type (ball or player) and team membership. The TOQ-Net outputs the action performed by the target player. Since there are only entity features, the input trajectory is encoded with a "unary" tensor of shape $T \times N \times D_1$, where $T$ is the length of the trajectory. That is, there are no nullary or binary inputs in this case.

## 3.2 RELATIONAL REASONING LAYERS

Our relational reasoning layers follow prior work on Neural Logic Machines (Dong et al., 2019), illustrated in Fig. 3 (i). Consider a specific time step $t$. At each layer $l$, the input to a neural logic layer is a 3-tuple $(P_{l-1}, Q_{l-1}, R_{l-1})$, which corresponds to nullary, unary, and binary features respectively. Their shapes are $D_0$, $N \times D_1$, and $N \times N \times D_2$. The output is another 3-tuple $(P_l, Q_l, R_l)$, given by

$$
\begin{aligned}
P_l &= \mathrm{NN}_P \left( \mathrm{Concat} \left[ P_{l-1}; \max(Q_{l-1}, \dim = 0) \right] \right), \\
Q_l &= \mathrm{NN}_Q \left( \mathrm{Concat} \left[ Q_{l-1}; \max(R_{l-1}, \dim = 0); \max(R_{l-1}, \dim = 1); \mathrm{expand}(P_{l-1}, \dim = 1) \right] \right), \\
R_l &= \mathrm{NN}_R \left( \mathrm{Concat} \left[ R_{l-1}; \mathrm{expand}(Q_{l-1}, \dim = 0); \mathrm{expand}(Q_{l-1}, \dim = 1) \right] \right),
\end{aligned}
$$

where $\mathrm{NN}_*$ are single fully-connected layers with sigmoid activations. For unary and binary features, $\mathrm{NN}_Q$ and $\mathrm{NN}_R$ are applied along the feature dimension. That is, we apply the same linear transformation to the unary features of all entities. A different linear transformation is applied to the binary features of all entity pairs. $\mathrm{Concat}[\cdot\,;\,\cdot]$ is the concatenation operation, applied to the last dimension of the tensors (the feature dimension). $\max$, also called the "reduced" max operation, takes the maximum value along the given axis of a tensor. The $\mathrm{expand}$ operation, also called "broadcast," will duplicate the input tensor $N$ times and stack them together along the given axis. The relational reasoning layer is applied identically to the input features at every time step $t$. That is, we use the same neural network weights in a relational reasoning layer for all time steps in parallel.

**Remark.** The design of relational reasoning layers is motivated by relational logic rules in a finite and fully grounded universe (i.e., under the closed-world assumption; see Dong et al. (2019) for a detailed discussion.) The $\max$ reduction operations implement a differentiable version of an existential quantifier over the finite universe of individuals, given that the truth values of the propositions are represented as values in $(0.0, 1.0)$. Because preceding and subsequent relational reasoning layers can negate propositions as needed, we omit explicit implementation of finite-domain universal quantification, although it could be added simply by including analogous $\min$ reduction operations. Thus, as illustrated in Fig. 3 (i), given input features $q_1(x, t)$ and $q_2(x, t)$, we can realize the logical formula $\exists x.\ q_1(x, t) \wedge q_2(x, t)$ by stacking two object reasoning layers.

Throughout the paper we have been using only nullary, unary, and binary features, but the proposed framework itself can be easily extended to higher-order relational features. From a graph network (Bruna et al., 2014; Kipf & Welling, 2017; Battaglia et al., 2018) point of view, one can treat these features as the node and edge embeddings of a fully-connected graph and the relational reasoning layers are instances of graph neural network layers.

## 3.3 TEMPORAL REASONING LAYERS

Temporal reasoning layers perform similar *quantification* operations as relational reasoning layers, but along the *temporal* dimension. They take the summarized event representation $P_K$ at each time step $t$ as input, produced by the $K$-th relational reasoning layer, as a matrix of shape $T \times D$. The output $P_{K+l}$ of a temporal reasoning layer $l$ is

$$
P_{K+l}^{(t)} = \max_{t' > t} \mathrm{NN}_l \left( \mathrm{Concat} \left[ \mathrm{Expand}(P_{K+l-1}^{(t')}, \dim = 1); \max_{t \le t'' < t'} P_{K+l-1}^{(t'')} \right] \right),
$$

where we use the superscript $P_{K+i}^{(t)}$ to denote the feature vector at time step $t$; $\mathrm{NN}_l$ is a single-layer fully connected neural network with sigmoidal output; $\max_{t \le t'' < t'} P_{K+l-1}^{(t'')}$ is a three-dimensional tensor indexed by $t$, $t'$, and $d$, where $d$ is the channel index.

| Model | Feat. | Reg. | Few-Shot | Full |
|---|---|---|---|---|
| STGCN | Raw | $73.1_{\pm1.3}$ | $28.5_{\pm4.7}$ | $63.2_{\pm0.5}$ |
| STGCN-MAX | Raw. | $72.9_{\pm1.4}$ | $25.1_{\pm5.9}$ | $62.3_{\pm0.7}$ |
| STGCN-LSTM | Raw | $74.1_{\pm1.4}$ | $25.7_{\pm6.4}$ | $63.3_{\pm0.8}$ |
| Space-Time | Raw. | $69.2_{\pm1.6}$ | $26.9_{\pm6.5}$ | $59.8_{\pm0.8}$ |
| Non-Local | Raw. | $69.5_{\pm1.4}$ | $44.1_{\pm7.7}$ | $63.9_{\pm1.2}$ |
| STGCN | Phy. | $73.2_{\pm1.6}$ | $26.0_{\pm5.7}$ | $62.8_{\pm0.6}$ |
| STGCN-MAX | Phy. | $73.6_{\pm1.5}$ | $28.6_{\pm5.0}$ | $63.6_{\pm0.7}$ |
| STGCN-LSTM | Phy. | $72.7_{\pm1.4}$ | $23.8_{\pm5.9}$ | $61.9_{\pm0.6}$ |
| Space-Time | Phy. | $74.8_{\pm1.5}$ | $31.7_{\pm6.1}$ | $65.2_{\pm0.6}$ |
| Non-Local | Phy. | $76.5_{\pm2.4}$ | $45.0_{\pm6.3}$ | $69.5_{\pm2.4}$ |
| TOQ-Net (ours) | Phy. | $\mathbf{87.7}_{\pm1.3}$ | $\mathbf{52.2}_{\pm6.3}$ | $\mathbf{79.8}_{\pm0.8}$ |

Table 1: Results on the soccer event dataset. Different columns correspond to different action sets (the regular, few-shot, and full action sets). The performance is measured by per-action (macro) accuracy, averaged over nine few-shot splits. The $\pm$ values indicate standard errors. TOQ-Net significantly outperforms all baseline methods on the few-shot action set.

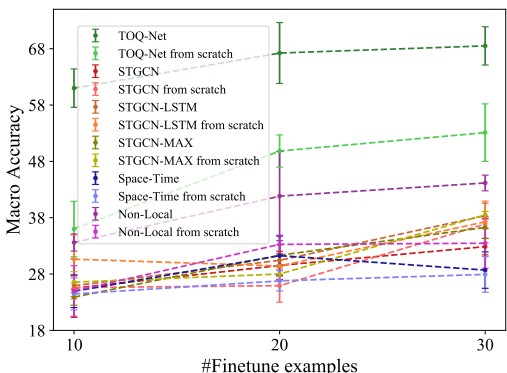

Figure 4: Generalization to soccer environments with a different court size and agent speeds. The standard errors are computed based on three random seeds.

**Remark.** The max operations act as existential quantifiers over time points. A max pooling of event feature $i$ from time $t$ to time $t' - 1$ evaluates to 1 iff event $i$ *eventually* happens within the time period. Since the preceding and subsequent fully connected networks can negate the proposition as needed, this operation can also represent universal quantification: event $i$ is *always* true from time $t$ to $t' - 1$. Note that since all temporal quantification is applied on the summarized state $P_K$, TOQ-Nets can represent *always* $\exists x.\ q_1(x) \wedge q_2(x)$, but not $\exists x.\ always\ q_1(x) \wedge q_2(x)$. This can be solved with interleaved relational and temporal quantification, but we find such interleaved models difficult to train and as yet out of scope of this paper.

## 4 EXPERIMENTS

We compare our model with other approaches to object-centric action recognition on two datasets The setups and metrics focus on data efficiency and generalization.

### 4.1 BASELINE

We compare TOQ-Nets against five baselines. The first two are spatial-temporal graph convolutional neural networks (STGAN; Yan et al., 2018) and its variant STGCN-MAX, which models entity relationships with graph neural networks, and temporal structure with temporal-domain convolutions. The third is STGCN-LSTM, which uses STGCN layers for entity relations but LSTM (Hochreiter & Schmidhuber, 1997) for temporal structures. The last two baselines are based on space-time graphs: Space-Time Graph (Wang & Gupta, 2018) and Non-Local networks (Wang et al., 2018). We provide details about our implementation and how we choose the model configurations in the appendix.

### 4.2 TRAJECTORY-BASED SOCCER EVENT DETECTION

We start our evaluation with an event detection task in soccer games. The task is to recognize the action performed by a specific player at specific time step in a soccer game, given a short game clip.

**Dataset and setup.** We collect training and evaluation datasets based on the gfootball simulator*, which provides a physics-based 3D football simulation. It also provides AI agents that can be used to generate random plays. The AI outputs abstract action commands: different types of passes (short pass, long pass, high pass, trap), ball shooting (shoot with the head or a foot), and defense (deflect, trip, slide, interfere, and catch). We run the simulator with AI-controlled players to generate plays, and formulate the task as classifying the action (9-way classification) of a specific player at a specific time step given a temporal context (25 frames). For all input training examples, we perform temporal alignment for the action so that the action happens in the middle of the clip. We randomly split all

---

*https://research-football.dev/

| Model | Feat. | 3v3 | 4v4 | 6v6 | 6v6 (Time Warp) | 8v8 | 11v11 |
|-------|-------|-----|-----|-----|-----------------|-----|-------|
| STGCN | Phy. | $40.7_{\pm1.0}$ (-40.4%) | $63.2_{\pm4.9}$ (-7.4%) | $68.2_{\pm2.8}$ (0.0%) | $52.8_{\pm7.0}$ (-22.6%) | $55.4_{\pm3.3}$ (-18.8%) | $44.4_{\pm2.1}$ (-34.9%) |
| STGCN-MAX | Phy. | $47.4_{\pm3.2}$ (-33.7%) | $68.8_{\pm2.0}$ (-3.8%) | $71.5_{\pm1.9}$ (0.0%) | $56.5_{\pm4.5}$ (-21.0%) | $59.1_{\pm0.7}$ (-17.3%) | $45.6_{\pm2.5}$ (-36.2%) |
| STGCN-LSTM | Phy. | $39.7_{\pm1.1}$ (-43.1%) | $60.4_{\pm0.2}$ (-13.5%) | $69.8_{\pm0.1}$ (0.0%) | $30.6_{\pm0.6}$ (-56.1%) | $55.8_{\pm2.0}$ (-20.0%) | $44.1_{\pm0.7}$ (-36.8%) |
| Space-Time | Phy. | $29.0_{\pm1.6}$ (-60.4%) | $53.5_{\pm3.2}$ (-27.0%) | $73.3_{\pm0.3}$ (0.0%) | $70.7_{\pm0.3}$ (-3.5%) | $33.9_{\pm2.8}$ (-53.7%) | $15.2_{\pm1.8}$ (-79.3%) |
| Non-Local | Phy. | $45.9_{\pm5.1}$ (-41.2%) | $70.7_{\pm5.3}$ (-9.5%) | $78.1_{\pm5.8}$ (0.0%) | $77.7_{\pm5.0}$ (-0.5%) | $58.5_{\pm10.8}$ (-25.1%) | $41.8_{\pm13.6}$ (-46.5%) |
| TOQ-Net | Phy. | $\mathbf{77.4}_{\pm3.5}$ (-12.4%) | $\mathbf{88.3}_{\pm0.7}$ (-0.0%) | $\mathbf{88.4}_{\pm0.6}$ (0.0%) | $\mathbf{86.9}_{\pm0.4}$ (-1.7%) | $\mathbf{81.3}_{\pm1.7}$ (-8.0%) | $\mathbf{77.1}_{\pm1.7}$ (-12.8%) |

Table 2: Results on generalization to scenarios with more agents and temporally warped trajectories on the soccer event dataset. The standard error of all values are smaller than 2.5%, computed based on three random seeds.

actions into two categories: seven "regular" actions, for which 3,000 video clips are available, and two "few-shot" actions, for which only 50 clips are available.

**Input features.** We implemented two types of input features. The first, *Raw*, uses the raw coordinate trajectories of all agents. The second, *Physics-based*, extracts the velocity and acceleration for each entity, as well as the distance between all pairs of agents. This pre-processing is necessary for TOQ-Nets, because they only model sequential structures of events, but not temporally local features such as velocity and acceleration. See the appendix for details.

**Results.** Table 1 shows the result. Our model significantly outperforms all the baselines in all three action settings. This suggests that our model is able to discover a set of useful features at both input and intermediate levels, which can be further used to compose classifiers for actions with only a few examples. In general, the models with physics-based features outperform those with raw coordinate inputs. Thus, for the rest of the experiments, we will use physics-based input feature for all models.

**Generalization to different input scales.** Intuitively, after learning how to detect events in a soccer game, the learner should also generalize concepts to analogs with the same basic structure but enacted on proportionately larger or smaller spatial or temporal scales—as in the soccer-inspired game futsal. To evaluate generalization to different input scales, we first train all models on the original dataset with 9-way classification supervision. After that, we finetune each model on a new dataset where all input values are doubled (i.e., the court is now double the size, and the players now move at four times the speed), but with only a small number of examples. Results are summarized in Fig. 4. While all baselines can benefit from the pre-training on the original dataset, our TOQ-Net outperforms all baselines by a significant margin when the finetuning dataset is small.

**Generalization to more players.** Due to their object-centric design, TOQ-Nets can generalize to soccer games with a varying number of agents. After training on 6v6 soccer games (i.e., 6 players on each team), we evaluate the performance of different models on games with different numbers of players: 3v3, 4v4, 8v8, and 11v11. Table 2 summarizes the result and the full results are provided in the appendix. Comparing the columns highlighted in yellow, we notice a significant performance drop for all baselines while TOQ-Net performs the best. By visualizing data and predictions, we found that misclassifications of instances of *shot* as *short pass* contribute most to the performance degradation of our model when we have more players. Specifically, the recall of *shot* drops from 97% to 60%. In soccer plays with many agents, a shot is usually unsuccessful and a player from another team steals the ball in the end. In such scenarios, TOQ-Net tends to misclassify such trajectories as a *short pass*. Ideally, this issue should be addressed by understanding actions based on agents' goals instead of the actual outcome (Intille & Bobick, 2001). We leave this extension as a future direction.

**Generalization to temporally warped trajectories.** Another crucial property of TOQ-Nets is to recognize actions based on their sequential order in the input trajectory, instead of binding features to specific time steps in the trajectory. To show this, we test the performance of different models on time warped trajectories. The results are shown in Table 2. Specifically, our test set consists of 25-frame trajectories, and the action may occur at anytime between the 6th and the 19th frame. By comparing rows with and without time warping, we notice a 60% performance drop for STGCN, STGCN-MAX, and STGCN-LSTM. In contrast, TOQ-Nets still have reasonable performance. Note that Space-Time

| Model | Reg. | 1-Shot | Full |
|---|---|---|---|
| STGCN | $99.89_{\pm 0.05}$ | $94.92_{\pm 1.03}$ | $98.79_{\pm 0.23}$ |
| STGCN-LSTM | $\mathbf{99.92}_{\pm 0.03}$ | $95.48_{\pm 1.67}$ | $\mathbf{98.86}_{\pm 0.43}$ |
| TOQ-Net | $\mathbf{99.96}_{\pm 0.02}$ | $\mathbf{98.04}_{\pm 0.97}$ | $\mathbf{99.48}_{\pm 0.24}$ |

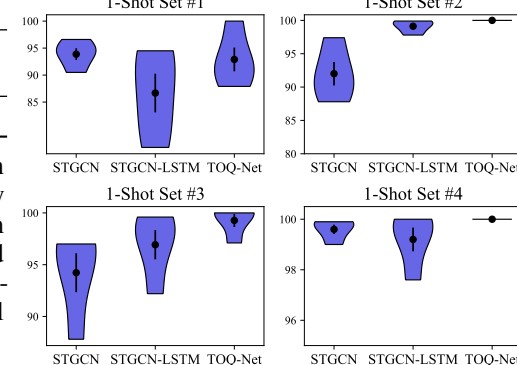

Table 3: Few-shot learning on the RLBench dataset, measured by per-action (macro) accuracy and averaged of four 1-shot splits and four random seeds per split. The ± values indicate standard errors. On the right we shows the sampled performance of different models on each individual 1-shot split.

and Non-Local model experienced almost no performance drop against time warping because they are completely agnostic to temporal ordering.

### 4.3 MANIPULATION CONCEPT LEARNING FROM 6D POSES

Structural action representations can also be usefully applied to other domains. Here we show the result in a robotic environment, where the goal is to classify the action performed by a robotic arm.

**Dataset and setup.** We generated a dataset based on the RLBench simulator (James et al., 2020), which contains a set of robotic object-manipulation actions in a tabletop environment. We chose 24 actions from the RLBench, including *opening box*, *closing fridge*, *push block*, etc. Details are provided in the appendix. For each object, at each time step, the input contains its bounding box (as an approximation of the shape of the object) and the 6D pose. Deformable objects are split into several pieces so that the learner can infer their state (open or closed). For example, the lid of the box is a separate object. We also input the 6D pose and the state of the robot gripper. During training, we used on average 65 videos for regular actions.

**Input features.** All methods take the raw trajectories as input. In TOQ-Nets, we use an extra single linear layer with sigmoid activation as the input feature extractor. To evaluate temporal modeling with recurrent neural networks, STGCN-LSTM uses kernel size 1 in its (STGCN) temporal convolution layers. Ablation studies can be found in the appendix.

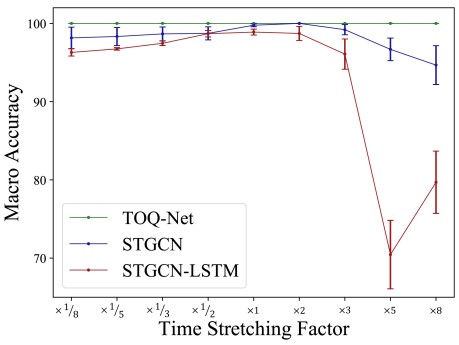

Figure 5: Comparing different models with different time stretching factors on the RL-Bench dataset.

**Results.** We start with evaluating different models on the standard 24-way classification task. We summarize results in Fig. 5. It also illustrates each model's performance under different time stretching factors of the input trajectory, from $1/8\times$ to $8\times$. Similar to the soccer dataset, the TOQ-Net outperforms both convolutional and recurrent models across all time stretching factors.

**Generalization to new action concepts.** We also evaluate how well different models can generalize the learned knowledge about opening and closing other objects to novel objects. Specifically, we hold out 50% of all *Open X* actions and 50% of the *Close X* actions to form the 1-shot action learning set. For example, the learner may only see a single instance of the opening of a box during training. Actions with more examples are "regular." All models are tested on a 3-way classification task: *Open-X*, *Close-X*, and *Other*. Results are summarized in Table 3. There are several held-out sets that are noticeably more "difficult" than others , such as the Set #1 in Table 3. Our visualization shows that actions *close jar* and *close drawer* are harder generalization targets, compared to other actions. This is possibly because the motion of *close jar* involves some rotation, and the motion of *close drawer* looks more like pushing than other *close-X* actions.

### 4.4 EXTENSION TO REAL-WORLD DATASETS

The proposed TOQ-Net can also be extended to other real-world datasets of human activities. We further conduct experiments on two datasets: Toyota Smarthome and a volleyball activity dataset.

**Toyota Smarthome (Das et al. (2019)).** Toyota Smarthome is a dataset that contains videos of daily living activities such as "walk", "take pills" and "use laptop". It also comes with 3D-skeleton detections. There are around 16.1k videos in the dataset, including 19 activity classes. The videos' length varies between a few seconds to 3 minutes and we subsample 30 frames for each video. We split them into training (9.9k), validation (2.5k), and testing (3.6k). Similar to the soccer event dataset, we treat the joints of the human as entities and use the physical feature extracted from the 3D-skeletons as the input. The input consists of the position of joints, the velocity of joints, the distance between pairs of joints, and joint angles. We evaluated our model and STGCN on the 19-way classification task. We also test model performance on time-warped sequences (2x quick motion).

|  | 19-way classification | Time warp (QuickMo 2x) |
|---|---|---|
| TOQ-Net | 42.0 | **41.2** |
| STGCN | **43.0** | 32.3 |

Our model achieves a comparable accuracy to STGCN on the standard classification task. Importantly, on the generalization test to time-warped sequences, our model encounters only a 0.8 performance drop where STGCN drops 10.7, which indicates that the learned temporal structures by our model improve model generalization to varying time courses.

**Volleyball Activity (Ibrahim et al. (2016)).** The volleyball dataset contains 4830 video clips collected from 55 youtube volleyball videos. They are labeled with 8 group activities such as "left spike" and "right pass". Each video contains 20 frames with the labeled group activity performed at the 10-th frame. The dataset also includes manual annotations for players at each frame, which consists of the bounding box, the indicator of whether the player is involved in the group activity, and the individual action such as "setting", "digging" and "spiking". We use the manual annotations, processed by an MLP as the input feature for each player. We train models to classify the video into one of the eight group activities. In our experiment, we adapt the train/val/test split in [2], i.e., 24, 15, and 16 of 55 videos are used for training, validation, and testing, respectively.

|  | 8-way classification | Time warp (Shift) | Time warp (QuickMo 2x) |
|---|---|---|---|
| TOQ-Net | 73.3 | **70.3** | **70.7** |
| STGCN | **73.6** | 39.5 | 48.6 |

On the standard classification task, TOQ-Net achieves a comparable performance with STGCN. When we perform time warping on the input sequences, STGCN's performance drops by >25.0% while our model drops only 3%. This further suggests the generalization ability of TOQ-Net w.r.t. varying time courses.

## 5 CONCLUSION AND DISCUSSION

We have presented TOQ-Nets, a neural module for describing complex activities by quantifying over both entities and time. TOQ-Nets use tensors to represent the time-varying properties and relations of different entities, and tensor pooling operations over different dimensions to realize temporal and object quantifiers. TOQ-Nets can learn to describe complex actions, and generalize well to scenarios with a varying number of entities and time courses.

The design of TOQ-Nets suggests multiple research directions. For example, the generalization of the acquired action concepts to novel objects, such as from *opening fridges* to *opening bottles*, needs further exploration. Meanwhile, TOQ-Nets are currently based on physical properties, e.g. 6D poses. Incorporating representations of mental variables such as goals, intentions, and beliefs can aid in action and event recognition (Baker et al., 2017; Zacks et al., 2001; Vallacher & Wegner, 1987). Finally, future research can examine connecting TOQ-Nets to perceptual processing, in order to individuate objects and compute their properties.

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

# A  APPENDIX

## A.1  SOCCER EVENT DATASET

We used the gfootball simulator[†] to generate simulated soccer match replays by letting the built-in AI play against each other. The simulator provides the 3D coordinates of the ball and the players as well as the action each player is performing at each time step. There are in total 13 actions defined in the simulator, including *movement, ball_control, trap, short_pass, long_pass, high_pass, header, shot, deflect, catch, interfere, trip* and *sliding*. We exclude *header* and *catch* actions, as they never appear in AI games. We also exclude *ball_control* and *movement*, since they just mean the agent is moving (with or without the ball). Thus, in total, we have nine action categories.

For each action, we have generated 5,000 videos, expect for *sliding*, for which we generated 4,000 videos because it is rare in the AI games. Among the generated examples, 60% (2,400 or 3,000) are used for training, 15% are used for validation, and 25% are used for testing.

Our data is collected from 6v6 soccer games. Each trajectory is an 8-fps replay clip that contains 17 frames (about two seconds). There is a single "target" player in each trajectory. The action label of the trajectory is the action performed by this target player at frame #9.

**Input.**  Each trajectory is represented with 7 time-varying unary predicates, including the 3D coordinate of each player and the ball and four extra predicates defining the type of each entity $x$: $\text{IsBall}(x)$, $\text{IsTargetPlayer}(x)$, $\text{SameTeam}(x)$, $\text{OpponentTeam}(x)$, where $\text{SameTeam}(x)$ and $\text{OpponentTeam}(x)$ indicates whether $x$ is of the same team as the target player. We also add a temporal indicator function which is a Gaussian function centered at frame of interest with variance $\sigma^2 = 25$.

**Generalization to different input scales.**  In this task, we multiply the coordinates of the ball and the agents by 2, and subsample the trajectories by a factor of 2. Thus, time flows twice fast, and the player moves four times the speed compared with the original dataset,

**Generalization to more players and temporally warped trajectories.**  In these tasks, we generate, on average, 1,500 examples for each action in 3v3, 4v4, 8v8, and 11v11 games. We use the same input and label format for non-time-warped trajectories as the original dataset. For time-warped trajectories, the trajectory length is 25, and each trajectory is labeled by the action performed by the target player at any time step between the 6-th and 19-th frame. We ensure that the target player performs only one action during the entire input trajectory. Thus, the label is unambiguous.

## A.2  RLBENCH DATSAET

We generated the RLBench dataset with the RLBench simulator (James et al., 2020). The simulator uses motion-planning algorithms that can generate robot arm trajectories that manipulate various objects in a tabletop environment. We use 24 actions from the dataset, including *CloseBox, Close-Drawer, CloseFridge, CloseGrill, CloseJar, CloseLaptopLid, CloseMicrowave, GetIceFromFridge, OpenBox, OpenFridge, OpenMicrowave, OpenWineBottle, PickUpCup, PressSwitch, PushButtons, PutGroceriesInCupboard, PutItemInDrawer, PutRubbishInBin, PutTrayInOven, ScoopWithSpatula, SetTheTable, SlideCabinetOpenAndPlaceCups, TakePlateOffColoredDishRack,* and *TakeToiletRollOffStand*. We randomly initialize the position and orientation of different objects in the scene and use the built-in motion planner to generate robot arm trajectories. Depending on the task and the initial poses of different objects, trajectories may have different lengths, varying from 30 to 1,000. Most of the trajectories have ~50 frames.

For each action category, we have generated 100 trajectories. Among the generated examples, 60% of the examples are used for training, 15% are used for validation, and the other 25% are used for testing.

**Input.**  Each trajectory contains the poses of different objects, at each time step. Each object except for the robot arm is represented as 13 unary predicates, including the 3D position, the 3D orientation

---

[†]https://research-football.dev/

represented as quaternions (4D vector), and the 3D bounding box (axis-aligned, 6D vector). The robot arm is represented as 8 unary predicates, including the 3D position, the 3D orientation, and a binary predicate that indicates whether the gripper is open or closed.

### A.3 IMPLEMENTATION DETAILS

In this section, we present the implementation detail of our model, the TOQ-Nets, and five baselines (STGCN, STGCN-LSTM, STGCN-MAX, Space-Time and Non-local), including the model architecture, input features, and the training methods.

#### A.3.1 TOQ-NETS

In the soccer event recognition task, we use three object quantification layers and three temporal quantification layers. Each object quantification layer has a hidden dimension of 16 (i.e., all nullary, unary, and binary tensors have the same hidden dimension of 16). Each temporal quantification layer has a hidden dimension of 48.

In the manipulation concept learning task, we use three object quantification layers and three temporal quantification layers. Each object quantification layer has a hidden dimension of 48. Each temporal quantification layer has a hidden dimension of 144.

**Input feature extractor.** We use the following physics-based input feature extractor for soccer event recognition. Given the trajectories of all entities, including all players and the ball, we first compute the following physical properties: ground speed (i.e., the velocity on the $xy$-plane), vertical speed (i.e., the velocity along the $z$ direction), and acceleration. We also compute the distance between each pair of entities.

After extracting the physical properties, we use the following feature templates to generate the input features. For each physical property, $X$, we first normalize $X$ across all training trajectories, and then we create $c = 5$ features:

$$\sigma\left(\frac{X - \theta_i}{\beta}\right),$$

where $\sigma$ is the sigmoid function, $\beta$ is a scalar hyperparameter, $\theta_i, i = 1, 2, 3, 4, 5$ are trainable parameters. These feature templates can be interpreted as differentiable implementations of $\mathbf{1}[X > \theta_i]$, where $\mathbf{1}[\cdot]$ is the indicator function. During training, we make $\beta$ decay exponentially from 1 to 0.001.

In the manipulation concept learning task, we use a single fully-connected layer with sigmoid activation as the feature extractor. The hidden dimension of the layer is 64.

#### A.3.2 STGCN

We use the same architecture as Yan et al. (Yan et al., 2018). Table 4 summarizes the hidden dimensions and the kernel sizes.

The STGCN model's output is a tensor of size $(T/4) \times N \times 256$, where $T$ is the length of the input trajectory, and $N$ is the number of entities. Following Yan et al. (Yan et al., 2018), we perform an average pooling over the temporal and the entity dimension and get a feature vector of size 256. We apply a softmax layer on top of the latent feature to classify the action.

**Input feature extractor.** In the soccer event recognition task, we have tested two different input features, namely the *Raw* feature and the *Physics-based* feature. For the *Raw* feature, the input to the STGCN model contains seven time-varying unary predicates (i.e., the 3D coordinates, and four predicates for entity types). For the *Physics-based* feature, in addition to the seven raw predicates, we also input the ground speed, the vertical speed, and the acceleration for each entity. Note that we do not input the distance between each pair of entities as we do for the TOQ-Nets, as the STGCN model does not support binary predicate inputs.

In the manipulation concept learning task, STGCN uses the same input format as the TOQ-Nets. Specifically, we represent the state of each entity with 13 unary predicates, including 3D position, 3D orientation, and bounding boxes.

| Model | Input Dim. | Output Dim. | Kernel Size | Stride | Dropout | Residual |
|---|---|---|---|---|---|---|
| | - | 64 | 7 | 1 | 0 | False |
| | 64 | 64 | 7 | 1 | 0.5 | True |
| | 64 | 64 | 7 | 1 | 0.5 | True |
| | 64 | 64 | 7 | 1 | 0.5 | True |
| STGCN | 64 | 128 | 7 | 2 | 0.5 | True |
| | 128 | 128 | 7 | 1 | 0.5 | True |
| | 128 | 128 | 7 | 1 | 0.5 | True |
| | 128 | 256 | 7 | 2 | 0.5 | True |
| | 256 | 256 | 7 | 1 | 0.5 | True |
| | 256 | 256 | 7 | 1 | 0 | True |
| | - | 16 | 7 | 1 | 0 | False |
| | 16 | 16 | 7 | 1 | 0.5 | True |
| $STGCN_S$ | 16 | 32 | 7 | 2 | 0.5 | True |
| | 32 | 32 | 7 | 1 | 0.5 | True |
| | 32 | 64 | 7 | 2 | 0.5 | True |
| | 64 | 128 | 7 | 1 | 0 | True |
| | - | 8 | 7 | 1 | 0 | False |
| $STGCN_T$ | 8 | 12 | 7 | 2 | 0.5 | True |
| | 12 | 64 | 7 | 1 | 0 | True |

Table 4: The STGCN architecture used in the paper.

| Model | Input Dim. | Output Dim. | Stride | Residual |
|---|---|---|---|---|
| | - | 64 | 1 | False |
| | 64 | 64 | 1 | True |
| | 64 | 64 | 1 | True |
| | 64 | 64 | 1 | True |
| Space-Time | 64 | 128 | 2 | True |
| | 128 | 128 | 1 | True |
| | 128 | 128 | 1 | True |
| | 128 | 256 | 2 | True |
| | 256 | 256 | 1 | True |
| | 256 | 256 | 1 | True |
| | - | 16 | 1 | False |
| | 16 | 16 | 1 | True |
| $Space\text{-}Time_S$ | 16 | 32 | 2 | True |
| | 32 | 64 | 2 | True |
| | 64 | 128 | 1 | True |
| | - | 32 | 1 | False |
| $Space\text{-}Time_T$ | 32 | 64 | 2 | True |
| | 64 | 64 | 2 | True |

Table 5: The Space-Time Region Graph architecture used in the paper.

### A.3.3 STGCN-MAX

The STGCN-MAX model is a variation of STGCN. During graph propogation, for each node, instead of taking average of all propagated information from all neighbors, we take their maximum value over all feature dimensions, and update the hidden state of each node with this. This baseline replicates a very similar propagation rule as our relational reasoning layer. However, it still uses temporal convolutions to model temporal structures.

| Model | Input Dim. | Output Dim. | Stride | Residual |
|---|---|---|---|---|
| Non-Local | - | 64 | 1 | False |
| | 64 | 64 | 1 | True |
| | 64 | 64 | 1 | True |
| | 64 | 64 | 1 | True |
| | 64 | 128 | 2 | True |
| | 128 | 128 | 1 | True |
| | 128 | 128 | 1 | True |
| | 128 | 256 | 2 | True |
| | 256 | 256 | 1 | True |
| | 256 | 256 | 1 | True |
| Non-Local$_S$ | - | 32 | 1 | False |
| | 32 | 32 | 2 | True |
| | 32 | 64 | 2 | True |
| | 64 | 128 | 1 | True |
| Non-Local$_T$ | - | 32 | 1 | False |
| | 32 | 32 | 2 | True |
| | 32 | 64 | 2 | True |

Table 6: The Non-local Neural Networks architecture used in the paper.

### A.3.4 STGCN-LSTM

The STGCN-LSTM model first encodes the input trajectory with an STGCN module. The output of the STGCN module is a tensor of shape $T \times N \times 256$, where $T$ is the length of the input trajectory, and $N$ is the number of entities. We perform an average pooling over the entity dimension and get a tensor of shape $T \times 256$. Next, we encode this feature with a 2-layer bidirectional LSTM module. The hidden dimension for the LSTM module is 256. We use a single softmax layer on top of the LSTM module to classify the action.

STGCN-LSTM uses the same architecture as STGCN for both tasks, except for the kernel size and the stride, because our goal is to evaluate the performance of recurrent neural networks (the LSTM model) on modeling temporal structures. In the soccer event recognition task, we use kernel size 3 and stride 1 so that the STGCN module will have the representational power to encode physical properties, such as velocity and acceleration. In the manipulation concept learning task, we use a temporal kernel size of 1 and stride 1. Meanwhile, Table 7 shows the ablation study of different kernel sizes, evaluated by the few-shot learning performance on the RLBench dataset. Different kernel sizes (1 and 3) have a similar performance.

### A.3.5 SPACE-TIME REGION GRAPH AND NON-LOCAL NEURAL NETWORKS

We use the same architecture for each Space-Time layer as Wang et al. proposed in Space-Time Graph (Wang & Gupta, 2018). Table 5 summarises the detailed parameters of all layers.

### A.3.6 NON-LOCAL NEURAL NETWORKS

We use the same architecture for each Non-local layer as Wang et al. proposed in Non-Local networks (Wang et al., 2018). The original non-local network works on pixels. Here instead, we apply it to the space-time graph constructed in the same way as Wang & Gupta (2018). The major difference between the Space-Time Graph and the Non-local Network is that Space-Time Graph uses graph convolutions while Non-local uses transformer-like attentions over all nodes in the graph. Table 6 summarises the detailed parameters of all layers.

| Model | Reg. | 1-Shot | Full |
|---|---|---|---|
| STGCN | $99.89_{\pm 0.05}$ | $94.92_{\pm 1.03}$ | $98.79_{\pm 0.23}$ |
| STGCN-LSTM (kernel size 3) | $99.89_{\pm 0.05}$ | $96.16_{\pm 1.39}$ | $99.04_{\pm 0.31}$ |
| STGCN-LSTM (kernel size 1) | $99.92_{\pm 0.03}$ | $95.48_{\pm 1.67}$ | $98.86_{\pm 0.43}$ |
| TOQ-Net | $\mathbf{99.96}_{\pm 0.02}$ | $\mathbf{98.04}_{\pm 0.97}$ | $\mathbf{99.48}_{\pm 0.24}$ |

Table 7: Few-shot learning on the RLBench dataset, measured by per-action (macro) accuracy and averaged of four 1-shot splits and four random seeds per split. The $\pm$ values indicate standard errors.

### A.3.7 TRAINING

**Soccer event recognition.** We use the identical training strategy for our model and both baselines. Each training batch contains 128 examples, which are sampled as following: we first uniformly sample an action category, and then uniformly sample a trajectory labeled as this action category.

In all the soccer event tasks, we optimized our model using Adam (Kingma & Ba, 2015), with the learning rate initialized to $\eta = 0.002$. The learning rate is fixed for the first 50 epochs. After that, we decrease the learning rate $\eta$ by a factor of 0.9 if the best validation loss has not been updated during the last 6 epochs.

All models are trained for 200 epochs. In the task of generalization to different input scales, we first train different models on the original dataset for 200 epochs ($\sim$40,000 iterations) and finetune them on the new dataset with few samples for 10,000 iterations.

**Manipulation concept learning.** We optimized different models using Adam with the initial learning rate $\eta = 0.003$. We applied the same learning rate schedule as in the soccer event recognition task. The batch size for the manipulation concept learning task is 32 for models with STGCN backbone and is 4 for TOQ-Nets. All models are trained for 150 epochs.

### A.4 STGCN-2D

The original STGCN does not support input features of higher dimensions, so we extend STGCN to STGCN-2D to add binary inputs such as the distance. This extension slightly improves the model performance on the standard 9-way classification test. It also helps in the few-shot setting. Specifically, on the few-shot learning setting:

| | Reg. | New | All |
|---|---|---|---|
| TOQ-Net | **87.7** | **52.2** | **79.8** |
| STGCN-2D | 84.1 | 46.9 | 75.8 |

Meanwhile, adding extra binary inputs does not improve the generalization of the network to games with a different number of agents or to time-warped sequences. This following table extends the Table 2 in the main text. Specifically, we train STGCN-2D on 6v6 soccer games and test it on scenarios with a different number of agents (3v3 and 11v11) and also temporally warped trajectories. Our model shows a significant advantage.

| | 9-way | 3v3 | 11v11 | 6v6(Time warp) |
|---|---|---|---|---|
| TOQ-Net | **88.4** | **77.4** | **77.1** | **86.9** |
| STGCN-2D | 84.5 | 17.5 | 16.6 | 39.7 |

### A.5 NETWORK ARCHITECTURE

In general, TOQ-Net has a smaller number of weights than baselines. We add additional comparisons to models of different #params. Specifically, $model_S$ is a smaller version of the model and $model_T$ is the tiny version of the model whose #params are about equal to or smaller than #params of TOQ-Net. To obtain smaller models we typically reduce the number of layers and the hidden state dimensions. On the few-shot task of the soccer event dataset, we test all baselines with all combination of features

| Model | Feat. | #params | Reg. | Few-Shot | Full |
|---|---|---|---|---|---|
| STGCN | Raw. | 3.43M | $71.3_{\pm 1.3}$ | $26.8_{\pm 4.6}$ | $61.4_{\pm 0.4}$ |
| **STGCN$_S$** | Raw. | 262K | $73.1_{\pm 1.3}$ | $28.5_{\pm 4.7}$ | **$63.2_{\pm 0.5}$** |
| STGCN$_T$ | Raw. | 34K | $68.7_{\pm 1.4}$ | $27.3_{\pm 5.7}$ | $59.5_{\pm 0.6}$ |
| STGCN | Phy. | 3.42M | $72.0_{\pm 1.5}$ | $22.6_{\pm 5.2}$ | $61.0_{\pm 0.8}$ |
| STGCN$_S$ | Phy. | 263K | $73.0_{\pm 1.5}$ | $22.5_{\pm 4.9}$ | $61.7_{\pm 0.5}$ |
| **STGCN$_T$** | Phy. | 34K | $73.2_{\pm 1.6}$ | $26.0_{\pm 5.7}$ | **$62.8_{\pm 0.6}$** |
| **STGCN-LSTM** | Raw. | 2.08M | $74.1_{\pm 1.4}$ | $25.7_{\pm 6.4}$ | **$63.3_{\pm 0.8}$** |
| STGCN-LSTM$_S$ | Raw. | 233K | $73.4_{\pm 1.4}$ | $24.2_{\pm 6.3}$ | $62.5_{\pm 0.6}$ |
| STGCN-LSTM$_T$ | Raw. | 33K | $71.9_{\pm 1.2}$ | $27.4_{\pm 6.6}$ | $62.0_{\pm 0.8}$ |
| STGCN-LSTM | Phy. | 2.08M | $72.5_{\pm 1.6}$ | $20.2_{\pm 5.0}$ | $60.9_{\pm 0.7}$ |
| STGCN-LSTM$_S$ | Phy. | 233K | $72.1_{\pm 1.7}$ | $20.5_{\pm 5.9}$ | $60.6_{\pm 0.7}$ |
| **STGCN-LSTM$_T$** | Phy. | 33K | $72.7_{\pm 1.4}$ | $23.8_{\pm 5.9}$ | **$61.9_{\pm 0.6}$** |
| STGCN-MAX | Raw. | 3.42M | $68.2_{\pm 3.9}$ | $17.9_{\pm 3.7}$ | $57.0_{\pm 3.3}$ |
| **STGCN-MAX$_S$** | Raw. | 262K | $72.9_{\pm 1.4}$ | $25.1_{\pm 5.9}$ | **$62.3_{\pm 0.7}$** |
| STGCN-MAX$_T$ | Raw. | 34K | $69.9_{\pm 1.2}$ | $29.1_{\pm 5.2}$ | $60.8_{\pm 1.0}$ |
| STGCN-MAX | Phy. | 3.42M | $73.5_{\pm 4.8}$ | $20.2_{\pm 6.0}$ | $61.7_{\pm 4.1}$ |
| **STGCN-MAX$_S$** | Phy. | 263K | $73.6_{\pm 1.5}$ | $28.6_{\pm 5.0}$ | **$63.6_{\pm 0.7}$** |
| STGCN-MAX$_T$ | Phy. | 34K | $74.5_{\pm 1.2}$ | $25.4_{\pm 6.1}$ | $63.6_{\pm 0.7}$ |
| Space-Time | Raw. | 263K | $67.9_{\pm 1.9}$ | $28.3_{\pm 7.0}$ | $59.1_{\pm 0.9}$ |
| **Space-Time$_S$** | Raw. | 24K | $69.2_{\pm 1.6}$ | $26.9_{\pm 6.5}$ | **$59.8_{\pm 0.8}$** |
| Space-Time$_T$ | Raw. | 13K | $68.0_{\pm 1.6}$ | $30.1_{\pm 6.7}$ | $59.6_{\pm 0.3}$ |
| Space-Time | Phy. | 263K | $71.3_{\pm 1.4}$ | $26.9_{\pm 7.1}$ | $61.4_{\pm 1.2}$ |
| Space-Time$_S$ | Phy. | 24K | $74.7_{\pm 1.4}$ | $30.7_{\pm 6.8}$ | $64.9_{\pm 0.6}$ |
| **Space-Time$_T$** | Phy. | 14K | $74.8_{\pm 1.5}$ | $31.7_{\pm 6.1}$ | **$65.2_{\pm 0.6}$** |
| Non-Local | Raw. | 1.23M | $53.2_{\pm 4.9}$ | $44.3_{\pm 7.5}$ | $51.3_{\pm 3.8}$ |
| **Non-Local$_S$** | Raw. | 107K | $69.5_{\pm 1.4}$ | $44.1_{\pm 7.7}$ | **$63.9_{\pm 1.2}$** |
| Non-Local$_T$ | Raw. | 32K | $63.8_{\pm 1.8}$ | $44.0_{\pm 6.2}$ | $59.4_{\pm 1.4}$ |
| Non-Local | Phy. | 1.23M | $74.5_{\pm 4.2}$ | $44.3_{\pm 7.0}$ | $67.8_{\pm 3.7}$ |
| **Non-Local$_S$** | Phy. | 108K | $76.5_{\pm 2.4}$ | $45.0_{\pm 6.3}$ | **$69.5_{\pm 2.4}$** |
| Non-Local$_T$ | Phy. | 32K | $75.6_{\pm 1.2}$ | $44.8_{\pm 6.4}$ | $68.8_{\pm 1.0}$ |
| TOQ-Net | Phy. | 35K | $87.7_{\pm 1.3}$ | $52.2_{\pm 6.3}$ | $79.8_{\pm 0.8}$ |

Table 8: Results on the soccer event dataset for baselines with different capacities, where model$_S$ and model$_T$ denote the small and the tiny variant for each model. The performance is measured by per-action (macro) accuracy, averaged over nine few-shot splits. The $\pm$ values indicate standard errors. For each baseline we showed the best performance over three levels of capacities. In fact, performances of most of the models are not affected much by the capacity. We highlight the best-performing variants of all baselines, and use them in all comparisons in the main text.

| Model | Feat. | Time Warp | 3v3 | 4v4 | 6v6 | 8v8 | 11v11 |
|-------|-------|-----------|-----|-----|-----|-----|-------|
| STGCN | Phy. | N | $40.7_{\pm1.0}$ (-40.4%) | $63.2_{\pm4.9}$ (-7.4%) | $68.2_{\pm2.8}$ (0.0%) | $55.4_{\pm3.3}$ (-18.8%) | $44.4_{\pm2.1}$ (-34.9%) |
| STGCN | Phy. | Y | $32.6_{\pm2.8}$ (-52.3%) | $50.2_{\pm4.4}$ (-26.5%) | $52.8_{\pm7.0}$ (-22.6%) | $43.2_{\pm4.9}$ (-36.7%) | $34.0_{\pm3.7}$ (-50.2%) |
| STGCN-MAX | Phy. | N | $47.4_{\pm3.2}$ (-33.7%) | $68.8_{\pm2.0}$ (-3.8%) | $71.5_{\pm1.9}$ (0.0%) | $59.1_{\pm0.7}$ (-17.3%) | $45.6_{\pm2.5}$ (-36.2%) |
| STGCN-MAX | Phy. | Y | $37.5_{\pm7.2}$ (-47.5%) | $52.3_{\pm4.2}$ (-26.9%) | $56.5_{\pm4.5}$ (-21.0%) | $46.6_{\pm3.7}$ (-34.8%) | $36.9_{\pm1.9}$ (-48.4%) |
| STGCN-LSTM | Phy. | N | $39.7_{\pm1.1}$ (-43.1%) | $60.4_{\pm0.2}$ (-13.5%) | $69.8_{\pm0.1}$ (0.0%) | $55.8_{\pm2.0}$ (-20.0%) | $44.1_{\pm0.7}$ (-36.8%) |
| STGCN-LSTM | Phy. | Y | $21.8_{\pm0.8}$ (-68.8%) | $27.8_{\pm1.3}$ (-60.2%) | $30.6_{\pm0.6}$ (-56.1%) | $25.8_{\pm1.0}$ (-63.1%) | $22.6_{\pm0.8}$ (-67.6%) |
| Space-Time | Phy. | N | $29.0_{\pm1.6}$ (-60.4%) | $53.5_{\pm3.2}$ (-27.0%) | $73.3_{\pm0.3}$ (0.0%) | $33.9_{\pm2.8}$ (-53.7%) | $15.2_{\pm1.8}$ (-79.3%) |
| Space-Time | Phy. | Y | $29.7_{\pm3.1}$ (-59.5%) | $51.6_{\pm2.5}$ (-29.6%) | $70.7_{\pm0.3}$ (-3.5%) | $33.8_{\pm2.2}$ (-53.9%) | $14.9_{\pm1.6}$ (-79.7%) |
| Non-Local | Phy. | N | $45.9_{\pm5.1}$ (-41.2%) | $70.7_{\pm5.3}$ (-9.5%) | $78.1_{\pm5.8}$ (0.0%) | $58.5_{\pm10.8}$ (-25.1%) | $41.8_{\pm13.6}$ (-46.5%) |
| Non-Local | Phy. | Y | $46.7_{\pm4.1}$ (-40.2%) | $69.9_{\pm4.7}$ (-10.5%) | $77.7_{\pm5.0}$ (-0.5%) | $58.7_{\pm12.6}$ (-24.9%) | $41.3_{\pm13.6}$ (-47.1%) |
| TOQ-Net | Phy. | N | $\mathbf{77.4}_{\pm3.5}$ (-12.4%) | $\mathbf{88.3}_{\pm0.7}$ (-0.0%) | $\mathbf{88.4}_{\pm0.6}$ (0.0%) | $81.3_{\pm1.7}$ (-8.0%) | $\mathbf{77.1}_{\pm1.7}$ (-12.8%) |
| TOQ-Net | Phy. | Y | $76.0_{\pm2.6}$ (-14.0%) | $87.7_{\pm1.4}$ (-0.8%) | $86.9_{\pm0.4}$ (-1.7%) | $80.3_{\pm1.1}$ (-9.1%) | $74.9_{\pm2.3}$ (-15.2%) |

Table 9: Full results on generalization to scenarios with all combinations of #agents and temporally warping on the soccer event dataset. The standard error of all values are smaller than 2.5%, computed based on three random seeds.

and capacities. Table 8 summarizes the results. In general, the "small" variations are the best for all models (except for STGCN-LSTM). Across all experiments presented in the main paper, we use the best architecture in Table 8.

### A.6 FULL RESULTS FOR THE TIME-WARPING GENERALIZATION

In the paper we have evaluated the generalization performance of different models to time-warped sequences but with the same number of players. Table 9 shows the results on time-warped sequences with different number of agents.

### A.7 #LAYERS OF TOQ-NET

We study the TOQ-Net performance over varying number of relation reasoning layers and temporal reasoning layers on the 9-way classification task of the soccer event dataset (See figure Fig. 6), and decide to use the combination of 3 relational layers and 4 temporal layers. Both relational layers and temporal layers play important roles in the performance.

### A.8 INTERPRETABILITY

Though interpretability is not the main focus of our paper, in Fig. 7, we show how TOQ-Net composes low-level concepts into high-level concepts: from inputs, to intermediate features at different temporal layers, and the final output labels. See the figure caption for detailed analysis.

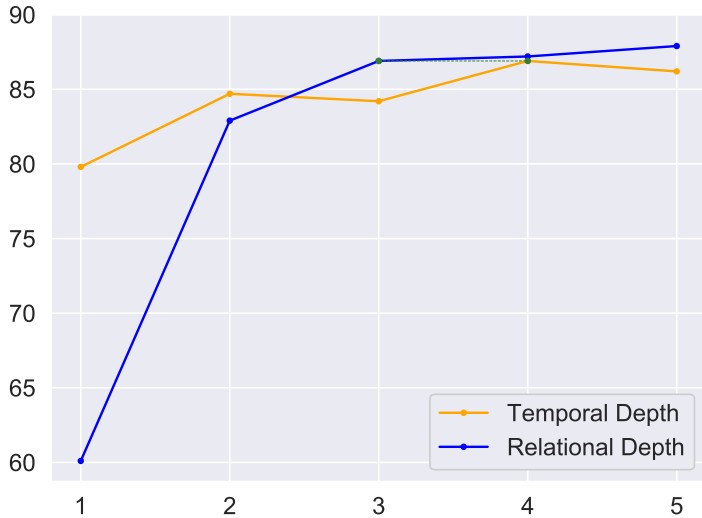

Figure 6: Comparing # of reasoning layers. When # of relational reasoning layers vary (blue), temporal reasoning layers are fixed at 4. When # of temporal reasoning layers vary (orange), relation reasoning layers are fixed at 3. Accuracy tested on 6v6 9-way classification. Adding reasoning layers improves results, and 3 relational layers + 4 temporal layers is a good balance between computation and performance.

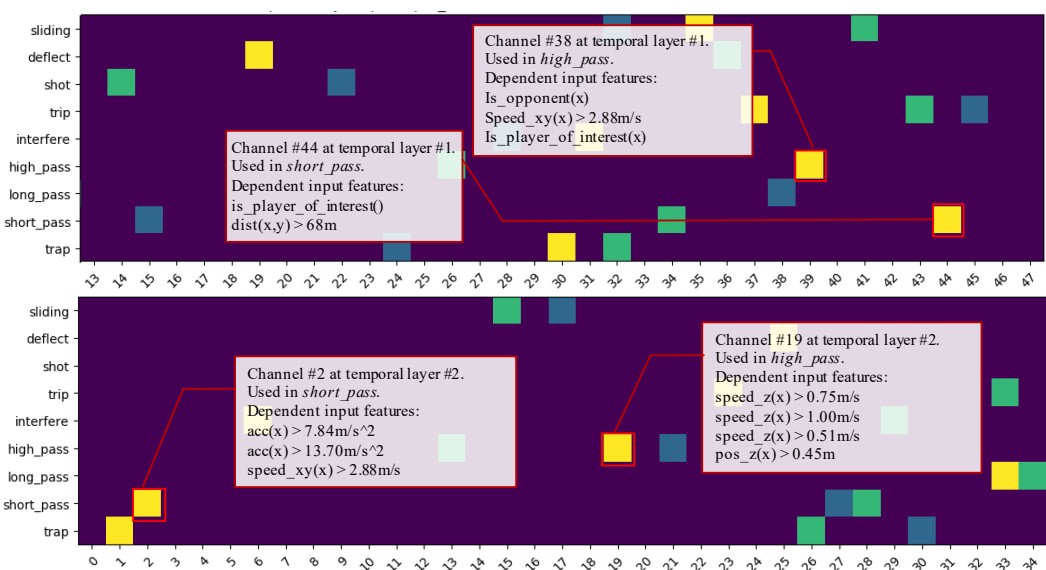

Figure 7: Relevant features in temporal layers. Feature dependencies are computed by gradient. These dependencies and thresholds are learned end-to-end from data. Insets detail features for events in the first and the second stage of the *high pass* and *short pass*.

