# OpenReview forum: "Temporal and Object Quantification Nets"
_ICLR.cc/2021/Conference — Reject_

### Official Review · AnonReviewer2 · 2020-10-15
**The proposed model is promising but the experiments are not sufficient**

**Rating:** 6
**Confidence:** 4

**Review:**

########################################################################

Summary:

This paper proposes TOQ-Nets which is a structured neural network that learns to describe complex activities over entities and time. The model leverages relational reasoning layers which are the Neural Logic Machines (NLM) to capture the spatial information. To further capture the temporal information, this paper proposes temporal reasoning layers.The results show that their method outperforms conventional graph neural networks with high accuracy and generalization with a large margin.

########################################################################

Pros:

- This paper extends the NLM into temporal. The proposed temporal reasoning layer borrows the idea from temporal logic

- programs which is interesting and promising.

- The Results show that their proposed model outperforms the STGCN with a large margin.

- The paper is well-organized and easy to read.

########################################################################

Although the proposed method is promising, the experiment settings are not convincing.

Cons:

- Missing reference : A line of research focuses on Group Activity Recognition [1, 2] which has the similar settings as Soccer Dataset. This paper collects the soccer dataset by themself and sets STGCN as their baseline which is not originally used for this task.

- Both datasets used in the paper are collected by the simulators. Compared with the Volleyball dataset [1, 2] that handles videos, the task in the paper seems much more simple and less practical.

- The inputs of TOQ-net and STGCN are different. The distances between players are not input to the STGCN, however the significance of this input variable is unknown. For example, the actions “Control ball” and “Movement (without ball)” can be easily distinguished by the distance of the target player and the ball. Please explain this.

- Missing Ablation study of the temporal reasoning layer. There is no experiment to show the improvement of the proposed temporal reasoning layer. Please explain how much important the temporal information is in your experiment.


##########################################################################

Reasons for score:

Overall, I like this idea of extending NLM into temporal. However, the experiments fail to convince me of its performance. So I tend to vote to reject temporarily. If the authors can show more evidence about the improvement of TOQ-net and eliminate my concerns, I will change my mind.

##########################################################################

Questions during rebuttal period:

Please address and clarify the cons above






##########################################################################

Reference

[1] A Hierarchical Deep Temporal Model for Group Activity Recognition. Mostafa S. Ibrahim et. al.

[2] stagNet: An Attentive Semantic RNN for Group Activity Recognition. Mengshi Qi et. al.


--------------------------------------------------------------------------------------------------------------------------------------------
I have read the rebuttal. The experiments in the rebuttals shows the effectiveness of TOQ-Nets in other large, real-world dataset. These experiments should be added in the revision of the paper and I would like to change my score to 6.

---

> ### Author Response · Authors · 2020-11-23
> **Our Response to Reviewer 2**
>
> Thank you very much for your constructive comments.
>
> **Q1: References.**
>
> A1: Thanks for the great suggestions. We have cited and discussed the suggested papers in our revision.
>
> - [1] A Hierarchical Deep Temporal Model for Group Activity Recognition. This paper has proposed a hierarchical deep temporal network that integrates a person-level and a group-level feature extractor. The high-level idea is to use RNNs to encode per-person features across the video and use another RNN to combine features of individual persons into a group feature for every frame. By contrast, our model uses relational reasoning layers to model interactions among agents for individual time steps and uses temporal reasoning layers to model temporal structures, such as the sequential orders of events.
>
> - [2] stagNet: An Attentive Semantic RNN for Group Activity Recognition. This paper combines graph neural networks with RNNs and spatial-temporal attention to model temporal information for individual nodes. Compared with theirs, we have proposed the temporal quantification operation for modeling temporal structures. Comparison between the proposed TOQ-Net and RNN-based representations, i.e., STGCN-RNN in the paper, for temporal structure modeling could be found in the paper. Our model outperforms RNN-based methods by a large margin on its generalization to time-warped sequences.
>
>
>
> **Q2: The inputs of TOQ-Net and STGCN are different.**
>
> A2: Please refer to our general response (Q1).
>
>
>
> **Q3: Evaluation of TOQ-Net on real-world datasets.**
>
> A3: Please refer to our general response (Q2).
>
>
>
> **Q4: Missing Ablation study of the temporal reasoning layer.**
>
> A4: We have included an ablation study on the effectiveness of relational/temporal reasoning layers in the appendix (Fig. 6). Specifically, we found that adding more temporal reasoning layers does help improve the overall accuracy. In the domains that we are considering, stacking 3 temporal reasoning layers achieves decent performance while adding more temporal layers shows little improvement. Informally, with 3 temporal reasoning layers, we can model events that can be decomposed into at most 3 sub-events.
>
> ```
> [1] A Hierarchical Deep Temporal Model for Group Activity Recognition. Mostafa S. Ibrahim et. al.
> [2] stagNet: An Attentive Semantic RNN for Group Activity Recognition. Mengshi Qi et. al.
> ```

---

### Official Review · AnonReviewer4 · 2020-10-24
**The paper proposes Temporal and Object Quantification Nets (TOQ-Nets), which can be used for learning composable action concepts from time sequences that describe the properties and relations of multiple entities. The authors test their model on two artificial benchmarks and demonstrate the effectiveness of their approach.**

**Rating:** 3
**Confidence:** 4

**Review:**

Strengths:
- The authors address a well formulated and an important problem for lots of practical scenarios.
- The paper is well written.
- The experiments demonstrate that the proposed method outperforms several baselines.

Weaknesses:
- I personally found the proposed approach to be very cumbersome and complicated. It consists of many different components that are not conceptually intuitive. Contrary to some of the recent approaches for modeling visual relations (i.e. Non-Local Networks or Space-Time Video Graphs), the proposed model is much more difficult to understand. In my opinion, this is a big disadvantage because the models that are most useful to the community are usually conceptually (and technically) simple, yet effective.
- The authors assume that the input features to the TOQ net are hand-engineered, and thus, are not learnable. This is very different to most modern approaches, which typically try to learn all the features from raw pixels end-to-end. Very few recent methods (to the best of my knowledge) rely on hand-engineered features for video modeling/action recognition. This begs the question how applicable the proposed approach is to modern computer vision community.
- The biggest weakness of the paper is its experimental evaluation. The authors evaluate their approach on 2 small-scale artificial video datasets. Thus, it is not clear whether the proposed approach would generalize to real datasets such as Kinetics, Something-Something, EPIC-Kitchens, etc. Most current action recognition methods evaluate their model on these large-scale real-world datasets. Thus, I think it is imperative that the authors would conduct thorough experiments not only on their small artificial datasets but also on the datasets that are most often used by the action recognition community. In particular, datasets like Something-Something, EPIC-Kitchens, or Charades require spatiotemporal relation modeling as demonstrated by prior work (i.e. Space-Time Video Graphs).
- The comparison with the pixel-level baselines (i.e. Non-Local Networks, or Space-Time Video Graphs) might not be exactly fair. The authors adapt these baselines to the dataset/task specific scenarios. Compared to their own model, the authors don't have as many incentives to tune these baseline models for their specific tasks. Thus, I believe that the experiments on the real-world large scale datasets such as Kinetics, Something-Something and Charades are essential for validating that the proposed approach is better than these prior methods.
- Missing relevant work: Wang et al., "Something-Else: Compositional Action Recognition with Spatial-Temporal Interaction Networks." (CVPR 2020).

Rebuttal Requests:
- The authors should include thorough experiments on the real-world large-scale datasets such as Kinetics, Something-Something, and Charades. The currently used datasets are not sufficient to verify the generalizability and usefulness of the proposed approach.

==Post Rebuttal Response:

I read the rebuttal, and unfortunately it didn't address most of my pressing concerns. I appreciate the authors' efforts to add new experiments on other datasets. However, in my opinion these new datasets are not very relevant for the action recognition community, i.e. they are small, and they are rarely used to compare the effectiveness of a particular model. In my initial review, I listed a few datasets that are most commonly used for action recognition comparisons. In my view, without comparisons on these more popular datasets it is very difficult to tell the real value of the proposed approach. If the authors could demonstrate close to state-of-the-art performance on those datasets I would be more convinced that the proposed approach is effective. Currently, most of the comparison are done w.r.t baselines that are implemented by the authors which is insufficient in my opinion. Therefore, I stand by my original recommendation of rejecting the paper.

---

> ### Author Response · Authors · 2020-11-23
> **Our Response to Reviewer 4**
>
> **Q1: The complexity of TOQ-Net**
>
> A1: We will make revisions to our model presentation for better clarity. For now, we kindly refer the reviewer to the overview paragraphs in Section 3, for a holistic view of the proposed model. The first remark in Section 3 also discusses the representational power of TOQ-Nets.
>
> TOQ-Net makes a clear, simple, meaningful contribution: using object and temporal quantification operations from first-order logic and linear temporal logic to model the relational and temporal relationships between entities and events.   These representational structures have been demonstrated to be useful for characterizing complex temporal events in a variety of domains [1, 2].
>
> Although we demonstrate it on some tasks that seem related to computer vision, the TOQ-Net is  task-agnostic, is fundamentally different from action recognition models in computer vision, and has unique advantages such as strong generalization to scenarios where there are more agents and objects than were present during training. The learned relational and temporal representations are also robust with respect to temporally warped sequences, and easily transfer to other prediction tasks in a similar domain.
>
> In our model, the input transformation layer can be seen as defining basic predicates and relations that will be useful for defining more complex concepts. The relational reasoning layers can build quantified first-order logic (FOL) expressions over these basic properties and relations, and can construct expressions that could be interpretable as “the player is close to the ball.” The temporal reasoning layer can represent temporal operations applied to these complex expressions, such as “the player is close to the ball until the ball moves with high speed,” inspired by linear temporal logic (LTL). The quantification operations in these two types of layers realize FOL and LTL quantifiers: for example, the existential quantifier in the sentence “there is a player such that the player is close to the ball,” and the until quantifier in the sentence “the player is close to the ball until the ball moves with high speed.”
>
> The closest structure to the TOQ-Net that can be achieved with traditional NN structures would be to make GNN’s with one node per entity per time step and then stack them into a fairly deep structure, and this is the idea of the Space-Time graph [3]. We have made comparisons to their approach and TOQ-Net outperforms it.
>
>
>
>
> **Q2: Hand-engineered features.**
>
> A2: All of the input features used in our experiments are physical features such as positions and velocities that are part of the state representation in the physics engines we use. These features are not specifically designed for our domains, but are general across sports analysis and robotic manipulation. We also kindly refer the reviewer to our additional results on the application of TOQ-Net on real-world datasets, including the Toyota Smarthome and the volleyball dataset, which use human positions and skeletons as features.
>
>
>
> **Q3: Evaluation on real-world datasets.**
>
> A3: Please refer to our general response (Q2).
>
>
>
> **Q4: Relevant work: Wang et al.**
>
> A4: We have cited and discussed the suggested paper in the revision. Their dataset is similar in spirit to the few-shot learning settings in the robotic manipulation dataset used in our paper. Their temporal model uses the non-local neural networks for gathering information across frames, and our model outperforms non-local neural networks, as illustrated in our experiments.
>
>
>
> ```
> [1] William Brendel, Alan Fern, and Sinisa Todorovic. "Probabilistic event logic for interval-based event recognition." In CVPR, 2011.
> [2] Georgios E. Fainekos, Antoine Girard, Hadas Kress-Gazit, and George J. Pappas. "Temporal logic motion planning for dynamic robots." Automatica 45, no. 2 (2009): 343-352.
> [3] Xiaolong Wang, and Abhinav Gupta. Videos as Space-Time Region Graphs. In ECCV, 2018.
> ```

---

### Official Review · AnonReviewer1 · 2020-10-28
**Interesting application of logic formulation in complex event understanding**

**Rating:** 6
**Confidence:** 4

**Review:**

This paper overall presents a model that defines the multi-person activities using logic expressions and uses neural logic models to generate recognitions and predictions over events. Specifically, the model follows the neural logic machines (Dong et al.) to define the operations in the networks. Authors demonstrate their model under two tasks, trajectory-based soccer event detection, and robot object manipulation event understanding. Using their self-generated datasets from simulators, the authors found that their model performs better than other baselines.

Paper strengths:

+ The logic forms of event understanding is a quite interesting direction, this paper could be a nice contribution towards that field.
+ The experimental results support authors' claim on the model

Potential cons:
1. There are not very unique modeling contributions in this paper. The model used is an application of neural logic machines.

2. Although I like the logical way of defining the events, it seems that the current refinement modules are quite similar to graph neural networks' message passing operations. Is the reason for the proposed model outperforming baselines that there are higher-order terms considered compared to baselines? What if a graph net/relation net with high-order interaction terms is used? Is the logical form of expressing the events really necessary (v.s. a natural language/semantic expression of events + graph nets + high-order terms)?

3. Two relevant and missing citations on soccer analysis and human interaction analysis [1,2].

[1] Large-Scale Analysis of Soccer Matches using Spatiotemporal Tracking Data

[2] Structure inference machines: Recurrent neural networks for analyzing relations in group activity recognition

---

> ### Author Response · Authors · 2020-11-23
> **Our Response to Reviewer 1**
>
> Thank you very much for your valuable comments.
>
> **Q1: The difference of TOQ-Net from graph neural networks (GNN) and neural logic machines (NLM).**
>
> A1: We will include additional discussion of these relationships in our revision. We agree that these models are conceptually similar to each other, and we will clarify our contribution along the following lines.
> 1. Our relational reasoning layer is a direct application of NLM. It is similar to applying GNNs per-frame, except that it handles high-order edges (hyperedges) and uses logic-inspired quantification operations. It can provably realize first-order Horn clauses, up to the model complexity (e.g., the number of layers and the hidden dimensions).
> 2. Temporal quantification is fundamentally different from existential or universal quantification in first-order logic. Intuitively, first-order quantifications are permutation-invariant, but temporal quantification needs to take the order of events into consideration. Thus, we draw inspiration from linear temporal logic (LTL) and introduce temporal quantification layers to match the model theory of LTL.
> 3. Our model is substantially different from other existing work on applying GNNs to spatial-temporal modeling. For example, in STGCN, temporal information is captured with temporal convolutions. In Space-Time Graphs, the same entities from adjacent frames are connected with edges. This fundamentally limits the generalization of such networks because convolutional kernels are not scale-invariant. The authors used graph convolution on the derived graph for modeling input sequences. We have compared our work to both approaches and show advantages over them, especially in generalization to situations with a different number of entities or to time-warped sequences.
>
>
>
>
> **Q2: Graph net/relation net with high-order interaction terms.**
>
> A2: Please refer to our general response (Q1).
>
>
>
> **Q3: References.**
>
> A3: Thanks for the great suggestions. We have cited and discussed the suggested papers in our revision.
>
> - [1] Large-Scale Analysis of Soccer Matches using Spatiotemporal Tracking Data. This paper has proposed a “role-based” representation that can be used for individual and team behavior analysis. The authors cast this problem as a minimum entropy data partitioning problem and apply the EM algorithm. Compared with their model, our model learns neural representations from inputs (positions, velocities, etc.) and uses object and temporal quantification operations to model the relational and temporal structures in the input sequences.
>
> - [2] Structure inference machines: Recurrent neural networks for analyzing relations in group activity recognition. This paper proposed a framework that integrates graph neural networks and RNNs. Specifically, RNNs are used in the “message aggregation” step in GNN propagation, so that the hidden state of each node captures the temporal information. Compared with them, we use temporal quantification operations to model the temporal structures. We have also compared with other RNN-based representations, i.e., STGCN-RNN in the paper. Our model outperforms RNN-based methods, especially in its generalization performance.
>
>
> ```
> [1] Large-Scale Analysis of Soccer Matches using Spatiotemporal Tracking Data. Bialkowski et al.
> [2] Structure inference machines: Recurrent neural networks for analyzing relations in group activity recognition. Deng et al.
> ```

---

### Author Response · Authors · 2020-11-23
**Our General Response**

We thank all reviewers for their constructive comments. Below we address shared concerns on evaluation. We have also answered specific questions in individual responses and have updated our paper accordingly.

**Q1: Graph networks with inputs of higher dimensions such as distances (R1, R2).**

A1: Thanks for the suggestion. We have included additional experiments for a more comprehensive comparison between TOQ-Net and STGCN. Adding binary inputs such as the distance between each pair of agents (namely, STGCN-2D) slightly improves the model performance on the standard 9-way classification test. It also helps in the few-shot setting. Specifically, on the few-shot learning setting:


|          | Reg    | New    | All    |
| -------- | ------ | ------ | ------ |
| TOQ-Net  | $\mathbf{87.7}$ | $\mathbf{52.2}$ | $\mathbf{79.8}$ |
| STGCN-2D | $84.1$ | $46.9$ | $75.8$ |

Meanwhile, adding extra binary inputs does not improve the generalization of the network to games with a different number of agents or to time-warped sequences. The following table extends Table 2 in the main text. Specifically, we train STGCN-2D on 6v6 soccer games and test it on scenarios with a different number of agents (3v3 and 11v11), and also temporally warped trajectories. Our model shows a significant advantage.

|         |9-way | 3v3  | 11v11 | 6v6(Time warp) |
| ------- | --- | ---- | ---- | -------------- |
| TOQ-Net | $\mathbf{88.4}$ | $\mathbf{77.4}$ | $\mathbf{77.1}$ |$\mathbf{86.9}$|
| STGCN-2D | $84.5$ | $17.5$ | $16.6$ |$39.7$|



**Q2: Evaluation on more complex and practical real-world datasets (R2, R4).**

A2: Thanks for the comments as well as the suggested datasets. We have included results on two more real-world datasets:

- The Toyota Smarthome dataset. [1]

  Toyota Smarthome is a dataset that contains videos of daily living activities such as “walk”, “take pills”, and “use laptop”. It also comes with 3D-skeleton detections. There are around 16.1k videos in the dataset, including 19 activity classes. The videos’ length varies between a few seconds to 3 minutes and we subsample 30 frames for each video. We split them into training (9.9k), validation (2.5k), and testing (3.6k). Similar to the soccer event dataset, we treat human joints  as entities, and use the physical feature extracted from the 3D-skeletons as the input. The input consists of the position of joints, the velocity of joints, the distance between pairs of joints, and joint angles. We evaluated our model and STGCN on the 19-way classification task. We also test model performance on time-warped sequences (2x quick motion).


  |         | 19-way classification | Time warp (QuickMo 2x) |
  | ------- | --------------------- | ---------------------- |
  | TOQ-Net | $42.0$                | $\mathbf{41.2}$                 |
  | STGCN   | $\mathbf{43.0}$                | $32.3$                 |

  Our model achieves a comparable accuracy to STGCN on the standard classification task. Importantly, on the generalization test to time-warped sequences, our model encounters only a 0.8 performance drop where STGCN drops 10.7, which indicates that the learned temporal structures by our model improve model generalization to varying time courses.

- The Volleyball dataset. [2]

  The volleyball dataset contains 4830 video clips collected from 55 youtube volleyball videos. They are labeled with 8 group activities such as “left spike” and “right pass”. Each video contains 20 frames with the labeled group activity performed at the 10-th frame. The dataset also includes manual annotations for players at each frame, which consists of the bounding box, the indicator of whether the player is involved in the group activity, and the individual action such as “setting”, “digging”, and “spiking”. We use manual annotations, processed by an MLP as the input feature for each player. We train models to classify the video into one of the eight group activities. In our experiment, we adapt the train/val/test split in [2], i.e., 24, 15, and 16 of 55 videos are used for training, validation, and testing, respectively.


  |         | 8-way classification | Time warp (Shift) | Time warp (QuickMo 2x) |
  | ------- | -------------------- | ----------------- | ---------------------- |
  | TOQ-Net | $73.3$               | $\mathbf{70.3}$            | $\mathbf{70.7}$                 |
  | STGCN   | $\mathbf{73.6}$               | $39.5$            | $48.6$                 |

  On the standard classification task, TOQ-Net achieves a comparable performance with STGCN. When we perform time warping on the input sequences, STGCN’s performance drops by >25.0% while our model drops only 3%. This further suggests the generalization ability of TOQ-Net w.r.t. varying time courses.



```
[1] Toyota Smarthome: Real-World Activities of Daily Living. Das et al.
[2] A Hierarchical Deep Temporal Model for Group Activity Recognition. Mostafa et al.
```

---

### Author Response · Authors · 2020-11-24
**Revision Note**

We thank all reviewers for their constructive and helpful comments. We have made revisions to our paper according to the comments from the reviewers. Specifically, we have

- Added new references suggested by the reviewers with discussions.
- Included additional results on two new dataset: Toyota Smarthome [1] and Volleyball Activities [2].
- Included new ablation studies about the baseline model STGCN.
We have also highlighted our revision in blue in the pdf.


```
[1] Toyota Smarthome: Real-World Activities of Daily Living. Das et al.
[2] A Hierarchical Deep Temporal Model for Group Activity Recognition. Mostafa et al.
```

---

### Decision · Program_Chairs · 2021-01-07
**Final Decision**

**Decision:**

Reject

**Comment:**

This paper presents work on temporal logic representations in neural networks.  The paper builds on work on Neural Logic Machines (Dong et al.), adding temporal quantification.  The main positives to the method are this contribution of the temporal reasoning layers (e.g. iii in Fig. 2).  This layer provides an interesting extension of existing literature in the area.

The main concerns raised by the reviewers were the following:
- Contribution of the paper over previous NLM techniques
- Relation to graph neural networks and other message passing techniques
- Use of hand-crafted initial features and resultant comparisons to baselines
- Empirical validation

After reading the authors' responses the reviewers reconsidered their positions and engaged in discussion.  While the reviewers appreciate the addition of temporal reasoning layers as an interesting contribution, there is still concern over the magnitude of this contribution and its effectiveness.

Additional points raised in the discussion include
- Lack of evaluation on standard, challenging datasets and comparisons to state of the art
- Ablation study in response (Fig. 6) does not consider absolute removal of temporal layers (main contribution).

Overall, while the paper does contribute an interesting inductive bias for learning with temporal data, the current evaluation is limited in terms of its effectiveness at the classification tasks in the experiments.  Based on the concerns raised in the initial reviews and subsequent discussion, it was determined that the paper is not ready for publication in ICLR.